# Temporal and Spatial Gene Expression Profile of Stroke Recovery Genes in Mice

**DOI:** 10.3390/genes14020454

**Published:** 2023-02-09

**Authors:** Jan Götz, Frederique Wieters, Veronika J. Fritz, Olivia Käsgen, Aref Kalantari, Gereon R. Fink, Markus Aswendt

**Affiliations:** 1Faculty of Medicine, University of Cologne, 50923 Cologne, Germany; 2Department of Neurology, University Hospital Cologne, 50931 Cologne, Germany; 3Cognitive Neuroscience, Institute of Neuroscience and Medicine (INM-3), Research Centre Juelich, 52425 Juelich, Germany

**Keywords:** behavior, recovery rate, grid walk, rotating beam test, qPCR, cAMP pathway

## Abstract

Stroke patients show some degree of spontaneous functional recovery, but this is not sufficient to prevent long-term disability. One promising approach is to characterize the dynamics of stroke recovery genes in the lesion and distant areas. We induced sensorimotor cortex lesions in adult C57BL/6J mice using photothrombosis and performed qPCR on selected brain areas at 14, 28, and 56 days post-stroke (P14-56). Based on the grid walk and rotating beam test, the mice were classified into two groups. The expression of cAMP pathway genes *Adora2a*, *Pde10a*, and *Drd2*, was higher in poor- compared to well-recovered mice in contralesional primary motor cortex (cl-MOp) at P14&56 and cl-thalamus (cl-TH), but lower in cl-striatum (cl-Str) at P14 and cl-primary somatosensory cortex (cl-SSp) at P28. Plasticity and axonal sprouting genes, *Lingo1* and *BDNF*, were decreased in cl-MOp at P14 and cl-Str at P28 and increased in cl-SSp at P28 and cl-Str at P14, respectively. In the cl-TH, *Lingo1* was increased, and *BDNF* decreased at P14. *Atrx*, also involved in axonal sprouting, was only increased in poor-recovered mice in cl-MOp at P28. The results underline the gene expression dynamics and spatial variability and challenge existing theories of restricted neural plasticity.

## 1. Introduction

Stroke is a leading cause of long-term disability and death worldwide [1]. During the first three to six months after a stroke, almost all stroke patients experience some degree of functional recovery [2]. Spontaneous (biological) recovery is best studied for motor deficits, i.e., improvements in a measurable behavior without external intervention or therapy [3], and accounts for 70–80% of maximal potential recovery [4]. However, to date, it remains challenging to explain why the clinical outcome is highly variable, with some patients showing very poor recovery while others show substantial functional improvement [3]. Further, the mechanisms that underlie spontaneous recovery and how they can be enhanced to promote the recovery of function remain to be elucidated.

The results from large genome initiatives collecting data from stroke patients worldwide highlight potential candidates, such as an intronic variant of a gene encoding PPP1R21, a regulatory subunit of protein phosphatase-1 [5], apolipoprotein E (APOE), and brain-derived neurotrophic factor (BDNF), to be involved in brain plasticity and strongly related to the functional outcome [6].

Compared to extensive human studies, transcriptomics studies in mice in the context of stroke remain scarce. These studies focused on the acute phase and the impact of aging or astrocytes with no apparent correlation with the recovery status or prediction thereof [7,8,9]. In a transcriptome study of the motor cortex after a cortico-striatal stroke in mice [10], we were the first to uncover distinct pathways in both the ipsilesional and contralesional primary motor cortex (il-MOp and cl-MOp), which significantly correlated with improved recovery. In particular, the cAMP signaling in the cl-MOp was involved with a selective reduction of *Adora2a* (adenosine receptor A2A), *Drd2* (dopamine receptor D2), and *Pde10a* (phosphodiesterase 10A) expression in recovered mice [10]. Little is known about the role of these genes in stroke recovery, especially in the chronic phase. In the acute phase, activation of the receptor ADORA2A reduces inflammatory cell infiltration [11], and activation of the receptor DRD2 by the drug sinomenine reduces neuroinflammation in astrocytes [12]. Deactivation of PDE10A activates a cascade in neurons to control neuronal survival and plasticity [13]. In a mouse study with striatal (MCAO model) and cortical infarcts (photothrombosis) and a survival time of 9 weeks, daily injections of a PDE10A inhibitor improved motor recovery in striatal but not cortical stroke [14]. These findings were complemented by another study investigating pharmacological PDE10A deactivation in an experimental model of striatal stroke (middle cerebral artery occlusion, MCAO model) that showed decreased infarct volume, brain edema, blood–brain barrier leakage, and disseminated neuronal injury [15]. Other growth factors and axonal sprouting-related genes have been proposed to be involved with stroke recovery. Similar to human studies, *BDNF* was found to be involved in synaptic plasticity, intrinsic neuronal excitability, and a good repair of brain damage [16]. *Atrx* (α-Thalassemia X-linked Intellectual Disability Syndrome) is involved in gene regulation at interphase and is required for post-stroke axonal sprouting in vivo [17]. In contrast, Lingo1 (leucine rich repeat and immunoglobin-like domain-containing protein 1) encodes a transmembrane protein, a functional component of the Nogo receptor signaling complex, resulting in the inhibition of axonal outgrowth [18]. While previous studies made no difference between mice with good or poor recovery, they highlight the therapeutic potential and further need for experiments targeting molecular pathways in specific brain regions to improve recovery.

We hypothesized that the expression of genes associated with spontaneous functional recovery is far more dynamic, i.e., not restricted to the early phase after stroke, as well as differently regulated in regions relevant for sensorimotor behavior near and far from the initial ischemic lesion.

## 2. Materials and Methods

### 2.1. Experimental Study Design and Animals

The study was performed under the ARRIVE and IMPROVE guidelines [19]. The animal experiments were conducted in compliance with European and national animal care laws and institutional guidelines and approved by the Landesamt für Natur, Umwelt und Verbraucherschutz North Rhine-Westphalia under animal protocols number 81-02.04.2019.A309, 84-02.04.2014.A305, and 84-02.04.2016.A461. The animals were housed in individually ventilated cages under a 12 h light/12 h darkness cycle with access to water and food ad libitum. In total, n = 80 adult male C57Bl/6J mice (age: 10–12 weeks, stock: #000664, The Jackson Laboratory, Bar Harbor, ME, USA) were used. The mice were allocated to experimental groups randomly and assigned a unique study ID. The experimenters were blinded against the experimental group during the data recording and the primary data analysis. The project and all related experimental data were managed using our in-house developed electronic research database/lab book [20]. The animals were habituated and trained to perform behavioral tests three times during the week before the stroke. The baseline behavior was recorded three days before stroke surgery, followed by repetitive tests at 1, 3, 7, 14, 21, 28, 35, 42, 49, and 56 days post-stroke. The first subgroup (n = 47) of mice was used for the qPCR experiments, which ended at 14 (n = 17), 28 (n = 11), or 56 (n = 19) days post stroke. The animals were perfused intracardially with ice-cold 40 mL of phosphate-buffered saline (PBS, Merck, Darmstadt, Germany). The brain regions (primary motor area (MOp), thalamus (TH), striatum (Str), and primary somatosensory area (SSp)) of all animals were manually dissected, snap-frozen in liquid nitrogen, and later stored at −80 °C. The second subgroup (n = 33) was used for the quantification of the lesion size and location using in vivo MRI at 7 days post-stroke and the same classification into good (n = 15) and poor (n = 18) recovery mice.

### 2.2. Photothrombosis

The photothrombosis protocol to induce cortical lesions was applied as reported previously [21] with a variation in the amount of injected photosensitive Rose Bengal and different intensities of laser radiation. Briefly, the mice were anesthetized with 3–4% Isoflurane in oxygen and placed in a stereotactic frame (#504926, WPI, Friedberg, Germany). The surface of the head was disinfected with povidone-iodine (Betaisodona, Mundipharma, Limburg, Germany), and an incision was made along the midline from the eye level to the neck (~1.5 cm). The periosteum was retracted, and the surface of the skull was cleaned with PBS and a cotton swab. A laser (MGL-FN-561 nm, CNI, Changchun, China), fixed on the stereotactic frame, was pointed on Bregma, and the laser spot was moved to the coordinates of the primary motor cortex (M/L: 2.00 mm and A/P: 0.00 mm). The animals received an intraperitoneal (i.p.) injection of 1000 μg of the photosensitive dye Rose Bengal (#A17053, Alfa Aesar, Karlsruhe, Germany). The dye was allowed to distribute in the whole organism for 5 min. Next, the laser was projected through the intact skull for 15 min with a laser intensity of 50 mW. The laser intensity was calibrated before the experiment using a power meter (PM121D, ThorLabs, Bergkirchen, Germany). Sham surgery was conducted following the same procedures, including the injection of 1000 μg Bengal Rose, but without radiation. The wound was closed, and the animals were allowed to recover in a pre-warmed heating chamber (V1200, MediHeat, Dalton, GA, USA). The analgesia included treatment with 1 mg/mL Tramadol (#100040, Grünenthal, Aachen, Germany) for 3 days before and after the surgery in the drinking water and an intraoperative i.p. injection of 4 mg/kg Caprofen (Rimadyl, Pfizer, Karlsruhe, Germany).

### 2.3. Behavioral Testing and Scoring

We used two different behavioral tests, the rotating beam and the grid walk test, which have been described in the literature to show robust sensorimotor deficits and recovery for the chronic phase after experimental stroke [22,23]. Motor coordination and balance were monitored using the rotating beam, a modification of the balance beam, as described elsewhere [22]. Briefly, the mice were trained to walk over a 120 cm long beam that rotated at 6 rpm. For four sequential trials (not one after the other), the time, distance, and the number of hindlimb drops were evaluated. The trial was stopped when the mouse dropped (which happens in rare cases in the first week after the stroke). The results from the four trials were further used for the group comparison. Next to the rotating beam experiment, the ability of the mice to grasp a thin wire of a metal mesh was assessed with the grid walk test [24]. The mice could freely explore for up to 5 min the 30 cm large metal square grid (mesh size 12.7 × 12.7 mm, 1.05 mm diameter), which was placed at 50 cm height (to prevent the mice from simply escaping the arena by jumping down on the table). We assessed the total number of footsteps and foot faults via video analysis. A step is counted as a foot fault when one of the following faults occur: the foot misses the grid and slips through the grid hole; the foot grabs the grid, but is not providing support and slips through the grid hole; or the animal is resting with the foot on the grid or over a grid hole, but the wrist is below the level of the grid. The ratio of foot faults was calculated as the number of foot faults divided by the (number of foot faults + non-foot fault steps) × 100.

Post-surgery monitoring included a visual inspection, weighing, and modified neurological deficit scoring (mNDS) every day and once a week from 7 days post-surgery. The mNDS was adapted from [25] and included measures of general deficits (appearance of the eyes and the fur, spontaneous movement, epileptic behavior) and focal deficits (body/forelimb symmetry, circling behavior, gait).

### 2.4. MRI Data Acquisition and Processing

MRI data acquisition was performed at the Max Planck Institute for Metabolic Research, Cologne, Germany, using a 94/20USR BioSpec (Bruker, BioSpin, Ettlingen, Germany) as described previously [26]. Briefly, each mouse was fully anesthetized with 2–3% Isoflurane and fixed in an animal holder. For a T2-weighted MRI (T2w-MRI), a high-resolution rapid acquisition with a relaxation enhancement (Turbo-RARE) scan with the following parameters was used: 256 × 256 matrix size with 28 slices (0.4 mm slice thickness, no gap), field of view (FOV) = 17.5 × 17.5 mm^2^, repetition time TR = 5500 ms, echo time TE = 32.5 ms, flip angle 90°. All the MRI data were acquired using ParaVision 6.0.1. and stored as raw data in Bruker format. T2w-MRI was converted into NIfTI format and processed using our in-house developed software AIDAmri v1.1 [27], which is available for download including a detailed manual from https://github.com/aswendtlab/AIDAmri (accessed on 20 January 2023). For the quantitative lesion quantification, individual lesion masks, i.e., visible as T2 hyperintense region, were semi-automatically segmented and transformed to the Allen Mouse Brain Reference atlas space for comparison across groups as described previously [27].

### 2.5. Sample Preparation and qPCR Protocol

According to the manufacturer manual, the Invitrogen TRIzol^TM^ Reagent was used for total RNA extraction (15596026, Thermo Scientific™, Waltham, MA, USA). Approximately 50–100 mg of brain tissue from every region was used. The quality and quantity of total RNA were determined by a Nanodrop (Thermo Scientific™) using 1 μL of RNA.

The High-Capacity cDNA Reverse Transcription Kit from Applied Biosystems^TM^, Waltham, MA, USA (4368814) was used to transcribe the total mRNA into single-stranded cDNA. A total of 500 ng of RNA diluted in 10 μL nuclease-free H20 were transcribed with a thermal cycler (Biometra, Analytika Jena, Jena, Germany), per the manufacturer’s protocol. The cDNA was stored at −20 °C. For the qPCR (LightCycler^®^ 96 System, Roche, Basel, Switzerland), according to the manufacturer’s instruction (Kapa SYBR Fast qPCR Master Mix, KR0389, Kapa Biosystems), the primer efficiencies were tested (Appendix A) using a random cDNA pool from 5 different samples (prepared as a concentration gradient with 0, 2.5, 5.0, 10.0, 50, 100, and 200 ng cDNA mix and triplets of 20 ng Mop, SSp, Str, and TH cDNA per well. Gapdh was used as the housekeeping gene, and 0 ng cDNA as the no-template control. The efficiencies and melting curves were calculated using the LightCycler^®^ 96 software.

### 2.6. Data Analysis and Statistics

Statistical analysis was performed using GraphPad Prism version 8.4.3 for macOS (GraphPad Software, San Diego, CA, USA).

### 2.7. Recovery Rate Analysis

The grid walk and rotating beam test data from days 3, 7, and 14 after stroke were used to dichotomize the mice according to their functional improvement. The linear regression between two (P3-14) and three time points (P3-7-14) was used as a surrogate of the “recovery rate” and calculated for the three readouts, speed, hindlimb drops, and foot faults individually (Appendix A). As an automated approach, cluster analysis was used to classify the mice based on possible similarities between the input data, using SPSS (Statistical Package for the Social Sciences). In the previous study from Ito et al., tMCAO was used instead of cortical photothrombosis, and the data from days 4, 8, and 14 post-stroke were used for hierarchical cluster analysis [10]. SPSS performed the following steps: agglomeration schedule and proximity matrix with a range of solutions between 2–4 under statistics. The plots are shown as a dendrogram and Ward’s method with squared Euclidean distance as a cluster method. The first cluster analysis, filled with data from the grid walk test (foot faults) and the rotating beam test (speed and hindlimb drop), showed clusters that were mixed with sham and stroke animals (Appendix A).

A negative recovery rate above the median in foot faults and hindlimb (HL) drops means less foot faults or -drops at P7 and P14, respectively, compared to P3 and, therefore, a good recovery. In contrast, it was the other way around for the rotating beam speed: a positive recovery rate above the median indicates a good recovery since the mice were faster at P7/14 than P3 (Appendix A).

Next, the median recovery rates (days P3, P7, and P14) were calculated and set as a reference for dividing the stroke mice into good and poor recovery. The mice classified in at least two of three readouts as good recovery were finally grouped as the good recovery group. Likewise, mice were grouped into a poor recovery group if at least two of the three readouts suggested a poor recovery. Finally, mice with individual readouts that showed conflicting results were grouped as neutral (Appendix A).

Nonlinear fitting was performed with the following parameters: one-phase decay with no special handling of outliers and least squares regression as the fitting method. The following statistical analysis was performed by mixed-effects analysis with post-hoc Dunnet’s and Šidák multiple comparisons test. The extra sum-of-squares F test was used to test if one fit adequately fits all the data points.

### 2.8. qPCR Analysis

The qPCR data were analyzed with the LightCycler^®^ 96 software. The program shows the Cq values of each sample. The Cq values were further processed with Microsoft Excel. SPSS Statistics (IBM, Version 27) was used to investigate any possible cluster appearing in the data. For statistical analysis and group-wise comparison of relative expression, the “Relative Expression Software Tool” (REST) was used [28]. The REST software compares the change in the relative expression (Ct) of the different samples (Ct) [29] as follows:ΔΔCt = ΔCt (treated sample) − ΔCt (untreated sample)

In detail, this calculates the change between the sequence of interest and the housekeeping (reference) gene, in this case, Gapdh. The mathematical model is based on the correction for exact PCR efficiencies and the mean crossing point deviation between sample group(s) and control group(s) [28]. Additionally, the expression ratio results are tested for significance by a randomization test. If the Ct-values from a stroke mouse were compared to sham values, the sham values were the control. In the case of stroke–stroke or sham–sham, either one stroke/sham was chosen as the control.

## 3. Results

### 3.1. Spontaneous Functional Recovery

The sensorimotor deficits after cortical stroke were monitored over 8 weeks using the grid walk and rotating beam test. In the grid walk test (Figure 1A), stroke mice showed a significant increase in foot faults during the acute phase (BL vs. P3 *p* < 0.001). This deficit remained significantly higher than the baseline, i.e., pre-stroke, up to 56 days post-stroke (BL vs. P56 *p* < 0.001). The number of foot faults decreased significantly from the maximum at 3 to 56 days (P3 vs. P56 *p* < 0.001) but remained significantly higher than in sham mice at each time point (*p* < 0.033–*p* < 0.001). In the rotating beam test (Figure 1B), stroke mice were significantly slower on day 3 compared to the baseline (*p* < 0.01). Between days 3 and 14, the speed increased significantly (*p* < 0.001) and later on remained at a pre-stroke level. The sham mice showed no significant change in speed in the acute phase, but demonstrated a significant increase between the baseline and 28 days (*p* < 0.01). In the same test, but with hindlimb drops as an additional readout, stroke and sham mice differed significantly in the acute phase (at day 3, *p* < 0.05). In the stroke group, the initially increased number of hindlimb drops up to day 3 decreased significantly in the following 3 weeks (*p* < 0.01) and then remained unchanged until day 56 (Figure 1C).

When plotted as individual lines, mice with potentially high (blue) and low recovery rate (red) could be visually delineated (Figure 1D–F).

The grid walk and rotating beam test data from days 3, 7, and 14 subdivided the mice according to their functional improvement (Figure 2). Linear regression between two (P3-14) and three time points (P3-7-14), respectively, was used as a surrogate of the “recovery rate” and calculated individually for the three readouts: speed, hindlimb drops, and foot faults (Appendix A). As a result, n = 13 mice were clustered into the good recovery, n = 9 into the poor recovery, and n = 3 animals with contradictory recovery rates in the neutral group, respectively. The neutral group was excluded from further analysis. The good and poorly recovered mice showed a very distinct recovery behavior. No single nonlinear fit represented both groups, good and poor, equally (*p* < 0.001). The mice in the good recovery group (blue) improved by 50% of the initial deficit at P3 during the first two weeks (*p* < 0.001) and gained significantly higher recovery rates at 14 days (*p* < 0.001). In contrast, mice in the poor recovery group (red) improved only by 25% of the initial deficit in the first 14 days (*p* < 0.001). Importantly, mice in the good recovery group showed a significant improvement at every time point compared to day 3 (*p* < 0.002–*p* < 0.001), whereas mice in the poor recovery group improved only until day 28 (*p* < 0.033–*p* < 0.001). To exclude a possible influence of initial lesion size and location on group classification, in vivo T2-weighted MRI was performed at day 7 post-stroke in a separate group of mice, which were also classified according to good and poor recovery (Figure 2B). Quantitative atlas-based analysis revealed that neither the lesion volume nor the lesion location differed significantly between the two groups (Figure 2B–D).

### 3.2. Gene Expression Analysis

Based on the classification into good and poor recovery, gene expression analysis was performed at 14, 28, and 56 days post-stroke (P14-56) in the stroke and sham mice in the ipsilesional (IL) and contralesional (CL) primary motor cortex (MOp), primary somatosensory cortex (SSp), striatum (STR), and thalamus (TH). To simplify the statistical report, the results contain the comparison between good and poorly recovered mice (Figure 3, Figure 4, Figure 5 and Figure 6) and sham mice (Appendix A). For completeness, we also performed the other statistical comparisons, i.e., between il- and cl-hemisphere of the same or different groups (Figure 3, Figure 4, Figure 5 and Figure 6, shown in gray).

#### 3.2.1. Primary Motor Cortex

*Adora2a:* Compared to sham mice, *Adora2a* expression was significantly increased in il-MOp and partly also cl-MOp in both recovery groups at 14 and 56 but not 28 days (*p* < 0.01–0.05). At 14 and 56 days, *Adora2a* expression was 4.5 and 2.3 times higher, respectively, in cl-MOp of poorly compared to well-recovered mice (*p* < 0.05).

*Pde10a:* At 14 and 56 days, *Pde10a* had an increased expression in cl-MOp of poorly recovered mice (*p* < 0.05 and *p* < 0.01, respectively) compared to sham mice and compared to the cl-Mop of well-recovered mice (2-fold, *p* < 0.05 and *p* < 0.01, respectively).

*Drd2:* At 14 days, *Drd2* expression was higher in cl-MOP of poorly recovered mice compared to sham and the cl-MOp of well-recovered mice (*p* < 0.01, 7.5 times higher). At P56, *Drd2* was significantly higher expressed in the il-MOp of good and the cl-MOp of poorly recovered mice compared to sham (*p* < 0.01). 

In contrast to *Adora2a*, *Pde10a*, and *Drd2*, we found no change in expression for *Lingo1*, *Atrx*, and *BDNF*, in most cases, compared to the sham group, e.g., *Lingo1* in il- and cl-MOp of poorly recovered mice (*p* < 0.001) at P14. The expression in cl-MOp of poorly recovered mice was also lower with a factor of 0.6 compared to the cl MOp of well-recovered mice (*p* < 0.05). At P56, *Lingo1* expression was increased by a factor of 1.2 in cl-MOp of poorly compared to well-recovered mice (*p* < 0.05).

*Atrx:* At P14, *Atrx* showed significantly less expression (*p* < 0.001) in the il MOp of poorly recovered mice compared to the control group. At 28 days post-stroke, *Atrx* had a lower expression (*p* < 0.05) with a factor of 0.7 in the cl MOp of well-recovered mice. *Atrx* in the cl MOp of poorly recovered mice was 1.5 times higher expressed (*p* < 0.01) compared to the cl MOp of well-recovered mice. 

*BDNF:* At P14, the il- and cl-MOp of poorly (*p* < 0.001) recovered mice had a decreased expression of *BDNF*, with a decreased expression of about 0.6 between the cl of good and poorly recovered mice (*p* < 0.05). 

#### 3.2.2. Primary Somatosensory Cortex

*Adora2a:* At P14, *Adora2a* was less expressed in poorly recovered mice in the il- and cl-SSp (*p* < 0.001 and *p* < 0.05), respectively. At P28, *Adora2a* expression was higher compared to sham in the cl-SSp of well-recovered mice (*p* < 0.05) and compared to poorly recovered mice (*p* < 0.01). 

*Pde10a:* At P14, the expression of *Pde10a* was significantly reduced in the il- and cl-SSp (*p* < 0.001 and *p* < 0.05) in the poor recovery group compared to the sham. At P28, only the cl-SSp of well-recovered mice had a higher expression compared to the sham (*p* < 0.05), and the cl-SSp of poorly recovered mice also showed a lower expression with a factor of 0.3 (*p* < 0.01). 

*Drd2:* At P28, *Drd2* expression was significantly higher in cl-SSp of well compared to poorly recovered mice (*p* < 0.01). At P56, *Drd2* expression was significantly higher in il-SSp of well- and poorly recovered mice (*p* < 0.001 and *p* < 0.05). 

*Lingo1:* At P28 and 56, *Lingo1* expression was significantly lower than sham in cl-SSp and il-SSp, respectively (*p* < 0.001 and *p* < 0.01). The cl-SSp of poorly recovered mice showed a 1.5 higher expression (*p* < 0.01) of *Lingo1* compared to cl-SSp of well-recovered mice at P28. 

*Atrx:* Differences in the expression of *Atrx* were detected in cl-SSp of both well- and poorly recovered mice compared to sham (*p* < 0.05) at P14, and cl-SSp of poorly recovered mice at P56, respectively. 

*BDNF:* At P14 and 28, *BDNF* expression was significantly increased and decreased, respectively, compared to sham (*p* < 0.05 and *p* < 0.01). At P28, *BDNF* expression was significantly lower in cl-SSp of well-recovered mice, with a 1.7-fold lower expression (*p* < 0.01) compared to poorly recovered mice.

#### 3.2.3. Striatum

*Adora2a:* At P14, *Adora2a* expression was significantly lower in the cl-Str in poorly recovered (*p* < 0.01) mice compared to sham and the cl-Str of well-recovered mice (*p* < 0.01) (Figure 5). At P56, *Adora2a* expression was significantly lower in cl-Str in well-recovered mice compared to sham (*p* < 0.05) and poorly recovered mice, respectively, by a factor of 1.7 (*p* < 0.01). 

*Pde10a:* At P14, *Pde10a* expression was significantly lower in cl-Str of poorly recovered mice compared to sham (*p* < 0.01) and the cl-Str of well-recovered mice. At P28, *Pde10a* expression was significantly lower in cl-Str of well-recovered mice compared to sham (*p* < 0.01) and the cl-STR of poorly recovered mice (2-fold, *p* < 0.05). At P56, the expression of *Pde10a* was significantly decreased in the il- and cl-Str, respectively, of well-recovered mice compared to sham (*p* < 0.05–*p* < 0.01). In the poor recovery group, *Pde10a* expression was higher in the cl-Str compared to sham (*p* < 0.01). Thus, *Pde10a* expression in cl-Str was 2.4-fold higher in poorly compared to well-recovered mice (*p* < 0.001).

*Drd2*: At P14, *Drd2* expression was significantly lower in the il- (*p* < 0.05) and cl-Str (*p* < 0.05 and *p* < 0.01) of poorly recovered mice compared to sham. *Drd2* expression was significantly lower in poorly compared to well-recovered mice (*p* < 0.01), which was reversed at P28 (*p* < 0.05). 

*Lingo1:* The expression of *Lingo1* dynamically changed in the cl-Str. At P14 days, it was significantly lower (*p* < 0.01), and at P28, it was higher (*p* < 0.001) in well- compared to poorly recovered mice. At P56, no difference was detected, although *Lingo1* was increased in both groups above sham levels (*p* < 0.01). In contrast, in the il-Str there was only a modestly lower expression of *Lingo1* in well- compared to poorly recovered mice (*p* < 0.05).

*Atrx:* The expression of *Atrx* was unchanged at P14 and 28. At P56, there was only a modest increase in *Atrx* expression in the il-Str of poorly compared to well-recovered mice (*p* < 0.05).

*BDNF:* The expression of *BDNF* in the cl-Str of well compared to poorly recovered mice was lower at P14 (*p* < 0.05) but higher at P28 and 56 (*p* < 0.01 and *p* < 0.05, respectively). 

#### 3.2.4. Thalamus

*Adora2a:* The expression of *Adora2a* in cl-TH was lower at P14 (*p* < 0.001), unchanged at P28, and higher at P56 (*p* < 0.05) in good compared to poorly recovered mice (Figure 6). Likewise, expression of *Adora2a* in cl-TH of well-recovered mice was significantly lower at P14 and higher at P56, compared to sham (*p* < 0.01 and *p* < 0.001, respectively). 

*Pde10a:* Similarly, *Pde10a* expression in cl-TH was lower at P14 and higher at P56, respectively, in well-recovered mice, compared to sham (*p* < 0.01 and *p* < 0.001) and poorly recovered mice (*p* < 0.001 and *p* < 0.01). In the il-TH, *Pde10a* expression was significantly higher in poorly recovered mice, compared to sham (*p* < 0.01) and well-recovered mice (*p* < 0.01).

*Drd2:* The expression of *Drd2* followed the same pattern: a lower expression in cl-TH of good vs. sham and poorly recovered mice at P14 (*p* < 0.001) and a higher expression at P56 (*p* < 0.001 and *p* < 0.01, respectively). 

*Lingo1: Lingo1* was 1.2-fold upregulated (*p* < 0.001) at P14 in the cl-TH of poorly compared to well-recovered mice. At P56, both sides of the TH in the good recovery group had a significantly lower expression of *Lingo1* (*p* < 0.01), and *Lingo1* in the il-TH was significantly lower compared to poorly recovered mice (*p* < 0.01).

*Atrx*: For *Atrx*, there was no significant difference between the recovery groups or sham. 

*BDNF:* The expression of *BDNG* in cl-TH was lower at P14 in poorly recovered compared to sham (*p* < 0.05) and well-recovered mice (*p* < 0.01), respectively. In contrast, at P56, the expression of *BDNF* in cl-TH was higher in poorly compared to well-recovered mice (*p* < 0.05). 

## 4. Discussion

We studied the expression levels of stroke recovery genes in mice with photothrombotic lesions in the sensorimotor cortex at 2, 4, and 8 weeks post-stroke. The results reveal a highly dynamic gene expression specific to the functional recovery rate. We identified previously unknown patterns of gene expression restricted to specific brain regions, which can serve as the basis for a novel way to design treatments targeting stroke recovery genes and improve functional recovery. 

### 4.1. Spontaneous Functional Recovery

To differentiate the effect of localized changes in gene expression on behavioral improvement after stroke, we classified mice with good and poor recovery using the grid walk and rotating beam test data obtained during the first two weeks, the period with the expected highest rate of spontaneous behavioral improvement [10] and brain plasticity [30]. The combination of the two behavioral tests was used to increase the specificity of the functional recovery measurement and compensate for differences in test sensitivity. Consistent with previous studies, mice with photothrombotic lesions in the sensorimotor cortex showed functional deficits for up to 8 weeks on the grid walk, which tests grasping and gait [31,32,33]. In contrast, on the rotating beam, which tests coordination, locomotion, and balance, the deficits compared to the sham were only detected in the acute to early subacute phase [21]. The different sensitivity might be related to the relevance of efficient sensory and motor information integration. The intact sensorimotor cortex is necessary for highly coordinated voluntary movements [34], e.g., navigating and holding with the paws onto the thin grid. In contrast, a lesion in the sensorimotor cortex is considered to be less relevant for locomotion on the beam, with a broader range of possibilities to compensate the paretic paw(s) by lower-order motor system components (e.g., basal ganglia, cerebellum, brain stem), i.e., regaining pre-stroke speed and preventing the hindlimb paw from dragging [35,36].

This is in line with a distal MCAO mouse model (cortex only), showing only moderate loss of speed on the rotating beam in the first week [37]. In mice with much larger cortico-striatal lesions (proximal MCAO model), i.e., affecting higher and lower-order sensorimotor regions simultaneously, the deficit on the rotating beam is much more substantial and long-lasting [10]. Thus, the different functional deficits are not related to the experimental model (MCAO vs. photothrombosis) but rather due to the lesion location.

Notably, the individual recovery rate was relatively independent of the type of test. Only 10% of the mice could not be assigned to the good or poor recovery groups. This finding suggests that more general recovery mechanisms exist. Unlike the unsupervised clustering method used in our previous MCAO study [10], here we had a much higher sample number. However, the data were incomplete because of difficulties with video recordings, especially at later time points. Typical attempts to handle missing values, i.e., filling in data (imputation) or ignoring the missing data (marginalization) [38], were unsuccessful (Appendix A). We cannot exclude the possibility that there are other recovery groups in addition to the well- and poorly recovery mice. A recent meta-analysis in a human cohort of over 400 stroke patients showed five groups based on the Fugl–Meyer motor upper extremity (FM-UE) score [39]. For the application of this modeling approach in mice, however, there is no validated neurological score available to date, which is sensitive for the detection of long-term deficits and robust against the differences in lesion location and compensation [40]. Future studies will benefit from using multiple behavior tests, such as the rotating beam and grid walk test, in combination with the cylinder and pasta test, which are less affected by compensation and include skilled paw movements [40]. Additionally, kinematic analysis can provide a more unbiased and precise evaluation of behavioral motifs [41], which might be better suited for determining recovery rates and recovery groups [42].

Similar to our previous study, in which the same stroke model was applied for photothrombotic cortical lesions in combination with the detailed quantification of lesion size and location [21], we found that the primary motor (MOp) cortex, followed by the primary somatosensory (SSp) cortex were affected in all mice to the largest extent. In contrast, other sensorimotor areas, e.g., thalamus (TH) and striatum (Str), were outside the ischemic area. In line with our previous MCAO study in mice with larger cortico-striatal lesions [10], the categorization of good and poor recovery was independent of lesion size and location.

In this study, only young (adult) male mice were used to obtain data which can be compared with previous studies on stroke recovery genes. We note that a more heterogeneous group of mice should be used in future studies to better represent stroke patients [43].

### 4.2. Temporal and Spatial Dynamics of Stroke Recovery Gene Expression

We asked whether stroke recovery gene expression follows a specific temporal and spatial profile. In contrast to previous recovery gene studies in rodent stroke models [7,8,9], our focus was on a more long-term scale, ranging from the early subacute and late subacute to the chronic phase [3]. As mice, compared to humans, recover much faster, with the highest improvements seen in the first 2–3 weeks compared to 3–6 months [30,44,45], the time points of 2, 4, and 8 weeks post-stroke were chosen. Moreover, this study is the first to compare gene expression and functional recovery in multiple regions directly involved in the generation of movement [46], i.e., primary motor and somatosensory cortex (MOp, SSp), striatum, and thalamus. These brain regions are highly interconnected and form the core of the motor system responsible for voluntary movement [47]. Our focus here in the discussion will be on the homotopic regions in the contralesional hemisphere. From human and rodent studies, we know that recruitment of contralateral areas is crucial for recovery, and this recruitment occurs mainly in large strokes when the reorganization of the ipsilesional areas no longer suffices for functional recovery [21,46,48,49]. This approach allowed us to extract group-averaged temporal gene expression profiles in comparing tissue of the contralesional hemisphere of good vs. poorly recovered mice (Figure 7). 

Our analysis revealed that both the timing and distance from the initial stroke lesion impacted the gene expression. There was, however, no continuity in stroke recovery gene expression in a particular area nor a direct correlation between gene expression and distance from the lesion area. *Adora2a*, *Pde10a*, and *Drd2* were more expressed in the cl-MOp of poorly recovered mice at 14 and 56 days. A mirrored gene expression change was seen in cl-Str/TH, with an initially lower/higher expression switching to higher/lower expression at 14 and 56 days. Interestingly, in cl-SSp, there was no significant change in *Adora2a*, *Pde10a*, and *Drd2* gene expression at 14 or 56 days, but there was lower expression of all three genes in poorly recovered mice. This data is partially in line with our previous MCAO RNA-Seq study, in which *Adora2a* was increased in cl-MOp but not cl-SSp or cl-Str, and the concept of *Pde10a* inhibition to improve functional recovery, i.e., lower *Pde10a* expression in well-recovered mice. In this context, it is interesting to note that changes in gene expression of *Pde10a* are not limited to the striatum with the highest natural abundance of *Pde10a* [50,51] but can also be observed in other brain areas. Different from changes in the other brain regions, however, *Pde10a* was most consistently increased between 14 and 56 days post-stroke in poorly recovered mice. However, a similar long-term change was only found for *BDNF* in the striatum, with a mirrored expression level, i.e., increased at 14 and decreased at 28–56 days in poorly compared to well-recovered mice. On the one hand, these results align with a previous study showing that increased levels of *Pde10a* led to lower levels of cAMP and/or cGMP, less CREB activation, and, finally, less BDNF [13]. On the other hand, PDE10A inhibition was shown to affect functional recovery in striatal but not cortical stroke [14]. Our results show that even in the absence of PDE10A deactivation/inhibition, the opposing *Pde10a*/*BDNF* expression appears in the striatum of spontaneously recovering mice. 

PDE10A inhibition stimulates per-infarct remodeling and pyramidal tract plasticity [15,52]. In our data, the differential expression of genes more directly related to promoting or inhibiting axonal sprouting, *Atrx*, and *Lingo1*, respectively [53,54], varied between the recovery groups. While *Atrx* was transiently increased in the poor recovery group at P28 in the MOp only, *Lingo1* expression was lower at P14 in MOp and P28 in Str. However, in most comparisons, *Lingo1* expression was higher in the poor recovery group at both subacute and chronic time points, suggesting more limited axonal sprouting concerning poorer functional recovery, as well as disturbed myelin repair [55]. Where the causal relationship for peri-infarct axonal sprouting for functional recovery has been shown by blocking ephrin-A5, a growth inhibitor in reactive astrocytes [33], such evidence is missing for the contralateral hemisphere. There are, however, multiple rodent and non-human primate studies showing the existence of axonal sprouting in the contralateral hemisphere, e.g., in homotopic cortical areas and corticospinal projections [56,57,58]. Axonal sprouting from the contralateral cortex is related to the lesion size, as it was shown by viral tracing and in vivo DTI studies [21,59].

Similar to *Pde10a*, *Adora2a* and *Drd2* belong to the cAMP signaling pathway, which is known to regulate synaptic plasticity and be involved in functional recovery after stroke [60,61]. In addition, ADORA2A and DRD2 receptors have been associated with the suppression of neuroinflammation through the modulation of microglia after stroke [11,62,63]. Our results highlight the complexity of *Adora2a* and *Drd2* gene expression, which can be higher in poorly recovered mice in cl-MOp in the sub-acute and chronic phase, but also lower in cl-SSp and cl-TH in the late subacute and chronic phase. These findings warrant consideration in future studies, extending previous acute pharmacological activation of ADORA2A and DRD2 [12,62] to induce long-term neuroinflammatory effects and functional improvements. Notably, we detected a selective lower expression of *Drd2* in poorly recovered mice in the cl-TH, but also an increased expression of *Drd2* in il- and cl-TH compared to sham mice. These results highlight the importance of Drd as a potential anti-inflammatory target [64], especially in preventing neurodegeneration [65], and might result in a lower level of secondary neurodegeneration after cortical stroke [21]. 

## 5. Conclusions

Gene therapies for cancer [66] have proven the revolutionary potential of targeted therapies. Whereas stroke and stroke recovery cannot be related to a single gene malfunction, for several proteins encoded by stroke recovery genes, promising therapies have been tested [3,4,5,6,10,11,12,13,14,16,18]. Our results highlight previously unstudied temporal dynamics in specific regions beyond the phase of increased plasticity that need to be considered in designing new therapies. In future studies, it will be essential to identify the cells that mainly express the stroke recovery genes and form functional proteins. This will allow for more specific characterization of the intercellular mechanisms and pharmacological or optogenetic studies to elucidate causal relationships with functional recovery.

A recent MCAO study in rats found Drd2 and Adora2a among the top differentially expressed genes in the contralesional hemisphere [67]. This suggests a common genetic cluster after stroke that is not restricted to mice and calls for more extensive cross-species comparisons.

## Figures and Tables

**Figure 1 genes-14-00454-f001:**
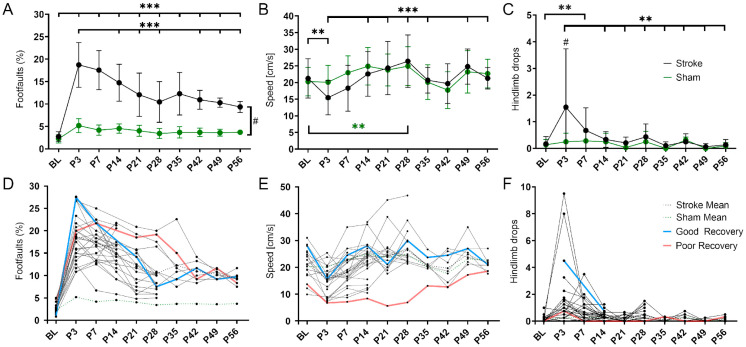
Variability in spontaneous functional recovery after cortical stroke. (**A**) The percentage of foot faults per step on the grid walk test was significantly increased in the stroke mice compared to the sham mice (*p* < 0.001) at all time points post-stroke (P3-56). Over time, the stroke mice made significantly fewer foot faults than on day 3 (P3; *p* < 0.001 except P7 and P35. (**B**) The speed achieved on the rotating beam was significantly slower in the stroke mice at P3 than BL and did not reach BL levels later. Over time, the stroke mice were significantly faster compared to day 3 (P3; *p* < 0.01 except P14 and P42). The speed of the sham mice was not decreased but increased significantly between BL and P28 (*p* < 0.01). (**C**) The stroke mice made significantly more hindlimb drops on the rotating beam only at P7 compared to BL (*p* < 0.001). Over time, the stroke mice made significantly fewer hindlimb drops than on day 3 (P3; *p* < 0.01 except P7). The sham mice did not make more or fewer hindlimb drops after stroke. (**D**–**F**) The same data but shown here as spaghetti plots: mean stroke data (blue, dotted), mean sham data (red, dotted), and single stroke data (black). Light blue and light red indicate an example of good and poor recovery mice, respectively. Significant differences between time points with *p* < 0.01 (**), *p* < 0.001 (***) and groups with *p* < 0.05 (#) were based on the mixed-effects analysis with post-hoc Dunnett’s multiple comparisons test. Stroke n = 30; sham n = 14. The data are shown as mean ± SD (**A**–**C**) or mean and single values (**D**–**F**).

**Figure 2 genes-14-00454-f002:**
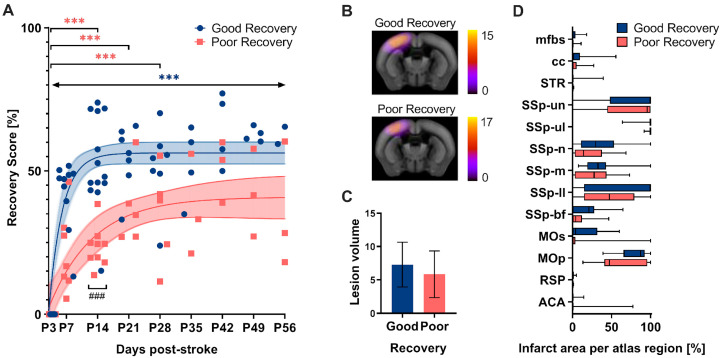
(**A**) The recovery score separated the stroke mice into groups with good and poor recovery. The recovery score at 3 days post-stroke (P3) was set to 0% (i.e., most significant deficit) with the baseline data neglected. At P14, the good recovery group (blue) reached approx. 50% recovery score, whereas the poor recovery group (red) reached only 25%. The good recovery group increased their recovery score over the entire period, while the poor recovery group reached a plateau. The recovery score improvement in the good recovery group significantly differed from P3 to P56 at every time point, and the poor recovery group differed selectively from P3 to P14, P21, and P28. The data are shown as individual points with nonlinear fits (thin line) and 95% confidence intervals. Significant differences between time points with *p* < 0.01, *p* < 0.001 (***) and groups with *p* < 0.001 (###), were based on the mixed-effects analysis with post-hoc Dunnett’s and Šidák multiple comparisons test, respectively; n = 13 (good recovery), n = 9 (poor recovery). (**B**) Incidence maps (generated from the average of individual stroke masks and overlay with the Allen Mouse Brain atlas) show the average lesion extent per group for a representative slice. (**C**) Quantification of lesion volume (mm^3^). (**D**) Atlas-based quantification of the percent infarct area per brain region: mfbs (medial forebrain bundle system), cc (corpus callosum), STR (striatum), SSp (Primary somatosensory area), un (unassigned), ul (upper limb), n (nose), m (mouth), ll (lower limb), bf (barrel field), Mos (Secondary motor area), MOp (Primary motor area), RSP (Retrosplenial area), ACA (Anterior cingulate area). The data are shown as mean/standard deviation and box plots. No significant differences between the groups based on *t*-test and 2-way ANOVA, respectively; n = 15 (good recovery), n = 17 (poor recovery).

**Figure 3 genes-14-00454-f003:**
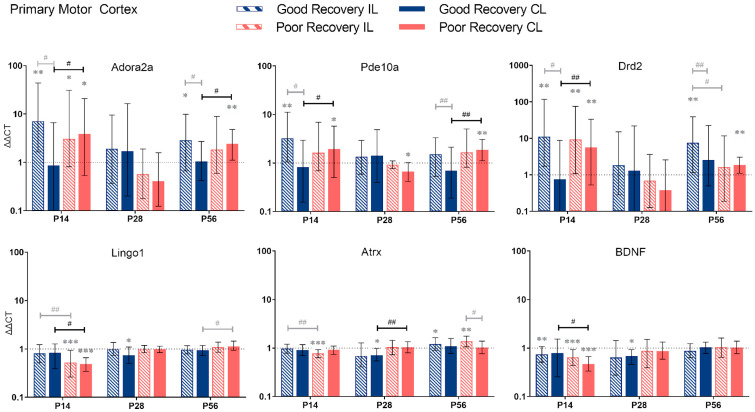
*Adora2a*, *Pde10a*, *Drd2*, *Lingo1*, *Atrx*, and *BDNF* gene expression in the primary motor cortex (MOp) at 14, 28, and 56 days post-stroke. Data shown as ΔΔCt (mean ± SE) compared to gene expression in sham mice and normalization to the expression of GAPDH as a reference gene, which was set at 1 (dotted line). Significant difference from sham shown as *p* < 0.05 (*), *p* < 0.01 (**), *p* < 0.001 (***). Significant differences between hemispheres and good/poor recovery, respectively, shown as *p* < 0.05 (#), *p* < 0.01 (##). The black lines indicated a significant comparison between contralesional good and poor. Data, expression, and significance calculated by REST software: P14: n = 5 good, 4 poor, 5 sham; P28: 3, 2, 3; P56: 4, 3, 5.

**Figure 4 genes-14-00454-f004:**
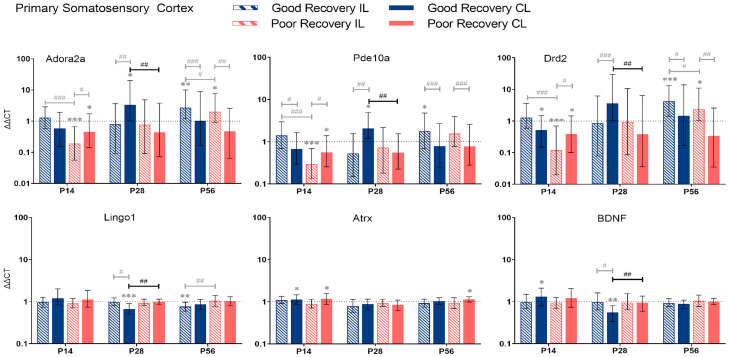
*Adora2a*, *Pde10a*, *Drd2*, *Lingo1*, *Atrx*, and *BDNF* gene expression in primary somatosensory cortex (SSp) at 14, 28, and 56 days post-stroke. Data shown as ΔΔCt (mean ± SE) compared to gene expression in sham mice and normalization to the expression of GAPDH as a reference gene, which was set at 1 (dotted line). Significant difference from sham shown as *p* < 0.05 (*), *p* < 0.01 (**), *p* < 0.001 (***). Significant differences between hemispheres and good/poor recovery, respectively, shown as *p* < 0.05 (#), *p* < 0.01 (##), *p* < 0.001 (###). The black lines indicate a significant comparison between contralesional good and poor. Data, expression, and significance calculated by REST software. P14 n = 5 good, 4 poor, 5 sham; P28: 3, 2, 3; P56: 4, 3, 5.

**Figure 5 genes-14-00454-f005:**
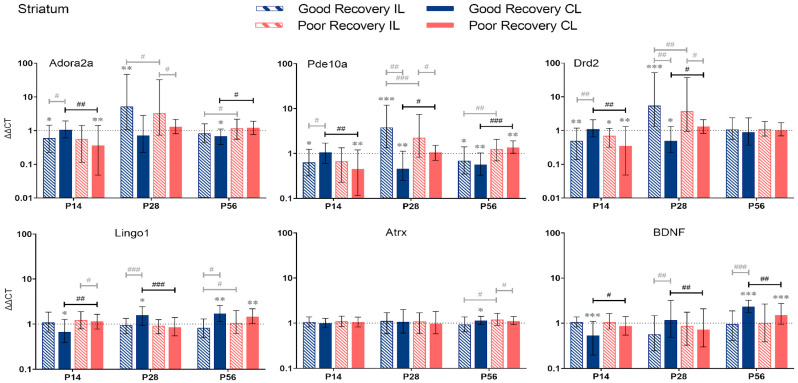
*Adora2a*, *Pde10a*, *Drd2*, *Lingo1*, *Atrx*, and *BDNF* gene expression in the striatum (Str) at 14, 28, and 56 days post-stroke. Data shown as ΔΔCt (mean ± SE) compared to gene expression in sham mice and normalization to the expression of GAPDH as a reference gene, which was set at 1 (dotted line). Significant difference from sham shown as *p* < 0.05 (*), *p* < 0.01 (**), *p* < 0.001 (***). Significant differences between hemispheres and good/poor recovery, respectively, shown as *p* < 0.05 (#), *p* < 0.01 (##), *p* < 0.001 (###). The black lines indicate a significant comparison between contralesional good and poor. Data, expression, and significance calculated by REST software: *p* < 0.05, *p* < 0.01, *p* < 0.001. P14 n = 5 good, 4 poor, 5 sham; P28: 3, 2, 3; P56: 4, 3, 5.

**Figure 6 genes-14-00454-f006:**
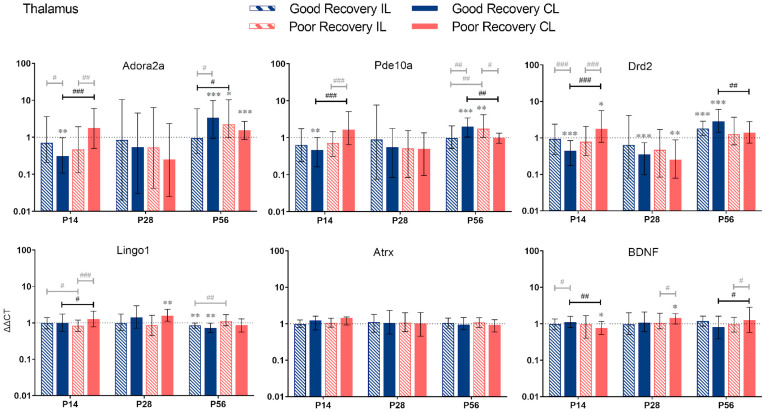
*Adora2a*, *Pde10a*, *Drd2*, *Lingo1*, *Atrx*, and *BDNF* gene expression in the thalamus (TH) at 14, 28, and 56 days post-stroke. Data are shown as ΔΔCt (mean ± SE), compared to gene expression in sham mice and normalization to expression of GAPDH as the reference gene, which was set at 1 (dotted line). Significant difference from sham shown as *p* < 0.05 (*), *p* < 0.01 (**), *p* < 0.001 (***). Significant differences between hemispheres and good/poor recovery, respectively, shown as *p* < 0.05 (#), *p* < 0.01 (##), *p* < 0.001 (###). The black lines indicate a significant comparison between contralesional good and poor. Data, expression, and significance calculated by REST software. P14: n = 5 good, 4 poor, 5 sham; P28: 3, 2, 3; P56: 4, 3, 5.

**Figure 7 genes-14-00454-f007:**
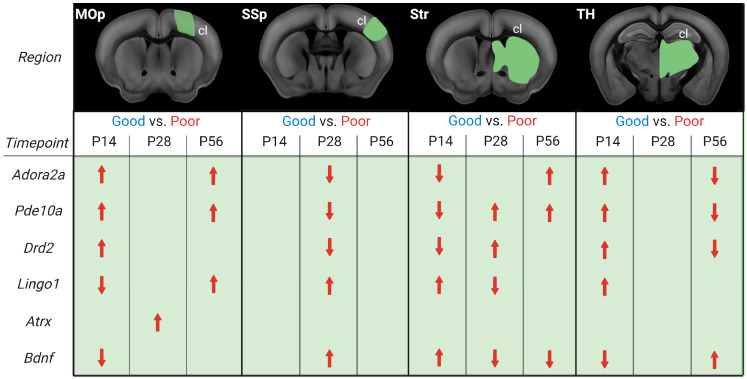
Illustration of differential gene expression of good and poorly recovered mice at 14, 28, and 56 days post-stroke. Gene expression analysis results of good compared to poorly (red) recovery mice, divided into contralesional (cl) brain parts and days post-stroke (P14, P28, P56). The red arrows indicate whether the gene’s expression was increased or decreased in poorly recovered mice compared to well-recovered mice. Figure created with BioRender.com (accessed on 28 December 2022).

## Data Availability

The raw and processed data comprising MRI, qPCR, and behavioral video recordings, as well as analysis sheets (Excel and GraphPad Prism) are available in the online repository GIN (https://doi.gin.g-node.org/10.12751/g-node.32p3ym/ (accessed on 20 January 2023)).

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
