# Peer review of "Temporal and Spatial Gene Expression Profile of Stroke Recovery Genes in Mice"

_genes, 2023, doi:10.3390/genes14020454_

Round 1
Reviewer 1 Report
Well written paper, exploring an important area of research.
Just a couple of comments:
The mice used were quite young, stroke is more common in the elderly, so maybe older mice may have been a better representative group?
How reliable it the photothrombsis method of inducing stroke, does the method induce stroke in the same regions and to the same extent? The size of the lesion and the areas affected may have impacted on how the mice were grouped into poor and good recovery. It might be hard to tell if mice in the good recovery were because the area of the initial infarct was smaller or affected different areas of the brain? A smaller infarct would suggest a better recovery rate and could impact qPCR results.
Figure 2 - the numbers of mice seem to drop in both poor and good recovery, this was due to poor video recordings? Did the mice numbers go down because they were being used for qPCR on days 14, 38 and 56? If so how did you choose the mice to use on these days for qPCR?
Overall very interesting paper.
Author Response
Reviewer 1:
Well written paper, exploring an important area of research.
Just a couple of comments:
The mice used were quite young, stroke is more common in the elderly, so maybe older mice may have been a better representative group?
Thank you for the positive feedback. We agree that the focus in experimental stroke research is mostly limited to young and healthy mice, which does not correspond all stroke patients. We consider it important to repeat the such experiments in a future study in both old and female mice. Since all the comparable studies used young mice, we decided to use mice in a similar age for our experiment to ensure the comparability of our study.
=> We have added this point in the Discussion Line 527 ff.
How reliable it the photothrombsis method of inducing stroke, does the method induce stroke in the same regions and to the same extent? The size of the lesion and the areas affected may have impacted on how the mice were grouped into poor and good recovery. It might be hard to tell if mice in the good recovery were because the area of the initial infarct was smaller or affected different areas of the brain? A smaller infarct would suggest a better recovery rate and could impact qPCR results.
Photothrombosis is known to be a very reliable minimal invasive method for lesioning specific parts of the brain. We and others have shown that there is a certain variability in lesion size, but the lesion location is very similar in all mice compared to other methods, i.e., MCAO (different to rats) produce larger variability [1–3].
We agree that it is important to verify the lesion size/location in this study. Due to the way we processed the tissue, i.e., dissection of multiple large brain areas, it was not possible to preserve sections for histology. We therefore decided to use a cohort of mice from a subsequent study with the same experimental paradigm but additional in vivo MRI at 7 days post stroke and detailed atlas-based lesion mapping as described in our previous study [3]. These results are in line with our MCAO RNAseq study, in which the different recovery groups showed also no difference in lesion size/location [4].
=> We have added the relevant text about in vivo MRI to Materials and Methods, Results and Discussion section including a new Figure 2B-D summarizing the results.
Figure 2 - the numbers of mice seem to drop in both poor and good recovery, this was due to poor video recordings? Did the mice numbers go down because they were being used for qPCR on days 14, 38 and 56? If so how did you choose the mice to use on these days for qPCR?
We started with one cohort of animals for qPCR, which received the same experimental paradigm but were sacrificed at different time points 14, 28, and 56. We initially aimed for a homogenous number of mice in the survival groups, but due to technical issues we were not able to obtain all data from the same number. The mice were randomly allocated to the different survival groups. After the experiment was done, the classification into good and poor recovery was done based on the behavioral data from all mice.
=> Details of the number of animals per survival group were added in Line 93 ff.
References
- Carmichael ST. Rodent models of focal stroke: Size, mechanism, and purpose. Neurorx. 2005;2: 396–409. doi:10.1602/neurorx.2.3.396
- Li H, Zhang N, Lin H-Y, Yu Y, Cai Q-Y, Ma L, et al. Histological, cellular and behavioral assessments of stroke outcomes after photothrombosis-induced ischemia in adult mice. Bmc Neurosci. 2014;15: 58. doi:10.1186/1471-2202-15-58
- Aswendt M, Pallast N, Wieters F, Baues M, Hoehn M, Fink GR. Lesion Size- and Location-Dependent Recruitment of Contralesional Thalamus and Motor Cortex Facilitates Recovery after Stroke in Mice. Transl Stroke Res. 2021;12: 87–97. doi:10.1007/s12975-020-00802-3
- Ito M, Aswendt M, Lee AG, Ishizaka S, Cao Z, Wang EH, et al. RNA-Sequencing Analysis Revealed a Distinct Motor Cortex Transcriptome in Spontaneously Recovered Mice After Stroke. Stroke. 2018;49: 2191–2199. doi:10.1161/STROKEAHA.118.021508
Reviewer 2 Report
The authors examined gene expression in C57Black/6J mice during the recovery process of cerebral infarction, dividing them into two groups: one with good functional recovery and the other with poor recovery. It is very interesting that gene expression of the cAMP pathway in chronic phase differed between the two groups in the present analysis by the authors. However, interpretation of differences in gene expression during the recovery process requires careful attention.
1) Please discuss how pathophysiology of cerebral infarction causes the prognosis of spontaneous functional recovery in the first two weeks after stroke? Is functional recovery correlated with voluntary movement? Is there any difference in cerebral blood flow in the acute phase of stroke? Is it a difference in the development of the collateral circulation in the subacute and chronic phase?
2) Please add figures to illustrate the regions of cerebral infarction and the difference in infarct size.
3) Basic physiological parameters should be described.
4) Please indicate how many male and female mice were used.
Author Response
Reviewer 2:
Please discuss how pathophysiology of cerebral infarction causes the prognosis of spontaneous functional recovery in the first two weeks after stroke?
The pathophysiology of stroke is triggered by a critical reduction of blood flow, which triggers a plethora of interacting mechanisms, i.e., neuronal cell death, edema formation, metabolic depression, neuroinflammation and astrogliosis. While these processes dominate the first days after stroke, there is a subsequent wave of plasticity mechanisms initiated, with a peak in the 2nd to 3rd week after stroke, which have direct influence on stroke recovery [1,2]. Our study goes beyond that concept and answers the question which long-term dynamics exist in the expression of single genes, which are known to be correlated with functional improvement.
Is functional recovery correlated with voluntary movement?
We did not assess the correlation of voluntary movement and functional recovery and to the best of our knowledge no mouse study has done that so far. Different to standardized clinical scales, it is technically complicated to measure comparable complex voluntary movements in mice. We have used established behavior tests to measure the improvement in natural movement patterns, e.g. walking, rearing, grasping, which are directly affected by the sensorimotor lesions. Voluntary in that context means that the mouse decides where, when and to which extent to perfom the movement without any external force. To answer your question for stroke patients: a study found that recovery of voluntary upper extremity movements in stroke patients cannot be predicted well from clinical variables (including e.g. lesion size, NIHSS score) [3].
Is there any difference in cerebral blood flow in the acute phase of stroke? Is it a difference in the development of the collateral circulation in the subacute and chronic phase?
Alterations in blood flow and vascular pathology after stroke have been described in detail in previous studies. In this study, the focus was not on the influence of the vascular system on recovery. Therefore, we did not include any related measurements. There is evidence that vascular remodeling is related to stroke recovery in patients [4]. A recent mouse study using longitudinal 2-photon imaging of the vascular network and blood flow provided strong evidence that vascular plasticity is a strong predictor for improved blood flow and functional recovery [5].
Whereas collateral blood flow is important to maintain neuronal viability until reperfusion [6], in photothrombotic stroke, there is no substantial collateral flow and reperfusion. Different to stroke models, i.e. middle cerebral artery occlusion, there is a localized vascular block leading to simultaneous vasogenic and cytogenic edema and non-reversible tissue death in a highly confined area with no or only very limited penumbra [7]. It should be further noted, that different to humans, a complete circe of Willis is present in only 10% of C57BL/6 mice [8] , suggesting a species-specific role of collateral flow for vascular plasticity.
Please add figures to illustrate the regions of cerebral infarction and the difference in infarct size.
We agree with the reviewer, that it is important to verify the lesion size/location in this study. Due to the way we processed the tissue, i.e., dissection of multiple large brain areas, it was not possible to perform histology in addition. We therefore decided to use a cohort of mice from a subsequent study with the same experimental paradigm but additional in vivo MRI at 7 days post stroke and detailed atlas-based lesion mapping as described in our previous study [9]. These results confirm our previous MCAO study, in which the different recovery groups showed also no difference in lesion size/location [10].
=> We have added the relevant text about in vivo MRI to Materials and Methods, Results and Discussion section including a new Figure 2B-D summarizing the results.
Basic physiological parameters should be described.
We did not measure basic physiological parameters, i.e. temperature, heart rate, respiration, as it is not part of our animal protocol approved by the authorities. Such measurements in mice are not easy to perform and require anaesthesia. We question the value of such measurements for the interpretation of our results in this study. What we have monitored quite extensively the impact of the surgery on the well-being of the mice according to a standardized score sheet, which measures various conditions including fur, ears, eyes, spontaneous activity, epilepsy, body symmetry, circling, fore- & hindpaw palsy, wounds, forepaw symmetry. There was no difference in the scoring between good and poor recovery mice.
Please indicate how many male and female mice were used.
We have used 80 male mice.
References
- Wieloch T, Nikolich K. Mechanisms of neural plasticity following brain injury. Curr Opin Neurobiol. 2006;16: 258–264. doi:10.1016/j.conb.2006.05.011
- Carmichael ST. The 3 Rs of Stroke Biology: Radial, Relayed, and Regenerative. Neurotherapeutics. 2016;13: 348–359. doi:10.1007/s13311-015-0408-0
- Koh C-L, Pan S-L, Jeng J-S, Chen B-B, Wang Y-H, Hsueh I-P, et al. Predicting Recovery of Voluntary Upper Extremity Movement in Subacute Stroke Patients with Severe Upper Extremity Paresis. Plos One. 2015;10: e0126857. doi:10.1371/journal.pone.0126857
- Durán-Laforet V, Fernández-López D, García-Culebras A, González-Hijón J, Moraga A, Palma-Tortosa S, et al. Delayed Effects of Acute Reperfusion on Vascular Remodeling and Late-Phase Functional Recovery After Stroke. Front Neurosci-switz. 2019;13: 767. doi:10.3389/fnins.2019.00767
- Williamson MR, Franzen RL, Fuertes CJA, Dunn AK, Drew MR, Jones TA. A Window of Vascular Plasticity Coupled to Behavioral Recovery after Stroke. J Neurosci. 2020;40: 7651–7667. doi:10.1523/jneurosci.1464-20.2020
- Jung S, Wiest R, Gralla J, McKinley R, Mattle H, Liebeskind D. Relevance of the cerebral collateral circulation in ischaemic stroke: time is brain, but collaterals set the pace. Swiss Med Wkly. 2017;147: w14538. doi:10.4414/smw.2017.14538
- Carmichael ST. Rodent models of focal stroke: Size, mechanism, and purpose. Neurorx. 2005;2: 396–409. doi:10.1602/neurorx.2.3.396
- McColl BW, Carswell HV, McCulloch J, Horsburgh K. Extension of cerebral hypoperfusion and ischaemic pathology beyond MCA territory after intraluminal filament occlusion in C57Bl/6J mice. Brain Res. 2004;997: 15–23. doi:10.1016/j.brainres.2003.10.028
- Aswendt M, Pallast N, Wieters F, Baues M, Hoehn M, Fink GR. Lesion Size- and Location-Dependent Recruitment of Contralesional Thalamus and Motor Cortex Facilitates Recovery after Stroke in Mice. Transl Stroke Res. 2021;12: 87–97. doi:10.1007/s12975-020-00802-3
- Ito M, Aswendt M, Lee AG, Ishizaka S, Cao Z, Wang EH, et al. RNA-Sequencing Analysis Revealed a Distinct Motor Cortex Transcriptome in Spontaneously Recovered Mice After Stroke. Stroke. 2018;49: 2191–2199. doi:10.1161/STROKEAHA.118.021508